# Characterization of Terpenoids in Aromatic Plants Using Raman Spectroscopy and Gas Chromatography–Mass Spectrometry (GC–MS)

**DOI:** 10.3390/ijms262311254

**Published:** 2025-11-21

**Authors:** Milagros Granda-Santos, Katherine Reyna-Gonzales, Llisela Torrejón-Valqui, Marvin G. Valle-Epquín, Aline C. Caetano, Jorge R. Díaz-Valderrama, Efraín M. Castro-Alayo, Ilse S. Cayo-Colca, Jorge L. Maicelo, César R. Balcázar-Zumaeta

**Affiliations:** 1Programa Doctoral en Ciencias para el Desarrollo Sustentable, Escuela de Posgrado, Universidad Nacional Toribio Rodríguez de Mendoza de Amazonas, Chachapoyas 01001, Peru; milagros.granda@untrm.edu.pe (M.G.-S.); icayo.fizab@untrm.edu.pe (I.S.C.-C.); 2Instituto de Investigación, Innovación y Desarrollo para el Sector Agrario y Agroindustrial (IIDAA), Facultad de Ingeniería y Ciencias Agrarias, Universidad Nacional Toribio Rodríguez de Mendoza de Amazonas, Chachapoyas 01001, Peru; 7702961920@untrm.edu.pe (K.R.-G.); llisela.torrejon@untrm.edu.pe (L.T.-V.); marvin.valle@untrm.edu.pe (M.G.V.-E.); efrain.castro@untrm.edu.pe (E.M.C.-A.); 3Instituto de Investigación para el Desarrollo Sustentable de Ceja de Selva, Universidad Nacional Toribio Rodríguez de Mendoza de Amazonas, Calle Universitaria N° 304, Chachapoyas 01001, Peru; aline.caetano@untrm.edu.pe (A.C.C.); jorge.diaz@untrm.edu.pe (J.R.D.-V.); 4Facultad de Ingeniería Zootecnista, Biotecnología, Agronegocios y Ciencia de Datos, Universidad Nacional Toribio Rodríguez de Mendoza de Amazonas, Chachapoyas 01001, Peru; jmaicelo@untrm.edu.pe

**Keywords:** bioactivity, essentials oils, herbal medicines, linalool, terpenoids

## Abstract

The study characterized the essential oils of eight aromatic medicinal plants (*Tagetes filifolia*, *Aloysia citrodora*, *Cymbopogon citratus*, *Eucalyptus globulus*, *Chamaemelum nobile*, *Piper aduncum*, *Minthostachys mollis*, and *Rosmarinus officinalis*) using Raman spectroscopy and gas chromatography–mass spectrometry (GC–MS). Raman spectra allowed the identification of bands associated with C–H, C=C, C–O, and C=O bonds, evidencing the presence of monoterpenes, sesquiterpenes, and oxygenated compounds. GC–MS analysis confirmed these results, detecting 224 compounds, predominantly terpenoids. Among the major compounds, cis,cis-nepetalactone (30.16%), β-caryophyllene (up to 18.26%), citronellol (10.92%), citral, and linalool were found. The combination of both techniques made it possible to relate the chemical composition to the molecular structure. This showed that the differences between species are mainly due to the proportion of oxygenated compounds (citral, linalool, geraniol) compared with aromatic hydrocarbons (β-caryophyllene, D-limonene, β-pinene). Additionally, the presence of cis,cis-nepetalactone in M. mollis was reported for the first time, representing a significant chemical finding.

## 1. Introduction

Currently, the food, pharmaceutical, and cosmetic industries show a growing interest in the use of natural compounds, driven by the demand for safer, more sustainable products with proven bioactive properties [1,2]. In this context, aromatic and medicinal plants represent a direct source of essential oils—volatile mixtures mainly composed of terpenoids, phenylpropanoids, aldehydes, and ketones—whose composition determines their biological activity [3,4]. These oils, obtained from leaves and stems through steam distillation, exhibit antimicrobial, antioxidant, anti-inflammatory, and insecticidal activities derived from the combined action of their secondary metabolites [5,6].

Several aromatic species cultivated between 2500 and 3500 m above sea level (a.s.l.), such as *Tagetes filifolia* (anise), *Aloysia citrodora* (lemon verbena), *Cymbopogon citratus* (lemongrass), *Eucalyptus globulus* (eucalyptus), *Piper aduncum* (matico), *Minthostachys mollis* (poleo), *Chamaemelum nobile* (chamomile), and *Rosmarinus officinalis* (rosemary), contain essential oils with specific chemical compositions dominated by monoterpenes, sesquiterpenes, and oxygenated compounds associated with their biological effects [7,8,9]. In addition, these species are traditionally used in folk medicine and gastronomy for their digestive, sedative, and anti-inflammatory properties, as well as for their ability to modify the aroma and flavor of foods [10,11,12,13,14,15].

The chemical characterization of these oils is essential to understand the relationship between their chemical composition and bioactive properties. A detailed analysis of the metabolites present in different species allows the identification of variations in chemical profiles that reflect genetic, environmental, and methodological differences, providing a framework for studying chemical diversity and its biological effects [16,17]. For chemical characterization, several complementary techniques have been employed, as recently reported in studies on essential oils, including confocal Raman spectroscopy [18], Ultra-high-performance liquid chromatography (UHPLC) [19], gas chromatography–mass spectrometry (GC–MS) [20], Fourier-transform infrared spectroscopy (FTIR) [21] and nuclear magnetic resonance (NMR) [22]. Confocal Raman spectroscopy is a rapid, non-destructive technique that provides direct spectral information relative to aromatic rings, enabling the identification of characteristic compounds such as terpenes, phenols, and others [23]. Likewise, GC–MS is used to identify and quantify volatile compounds, allowing the verification of results obtained through Raman spectroscopy [24,25,26,27]. The integration of both techniques therefore provides a more comprehensive view of the chemical composition, as Raman spectroscopy offers structural information that can be confirmed by GC–MS, which reveals the identity and relative abundance of the components [28,29]. This dual analytical approach enhances the robustness of chemical characterization by integrating information on compound identity, abundance, and structural features within essential oils. Moreover, it enables the establishment of correlations between chemical composition and the bioactive properties of the food matrix [29,30].

In this context, the present study aimed to characterize the chemical and structural composition of essential oils obtained from eight aromatic medicinal plants using Raman spectroscopy and GC–MS analyses. These techniques were complemented by multivariate analyses to group the predominant terpenoids according to the compounds shared among species, which are previously known for their medical, cosmetic, and industrial relevance.

## 2. Results and Discussion

### 2.1. Analysis of Raman Spectral Data

Raman spectroscopic analysis allowed the identification of the main vibrational bands associated with the predominant compounds present in the EO of *T. filifolia*, *A. citrodora*, *E. globulus*, *C. citratus*, *C. nobile*, *P. aduncum*, *M. mollis*, and *R. officinalis* (Figure 1 and Appendix A). The observed signals, reported here for the first time, reflected vibrational modes characteristic of C–H, C=C, C–O, and C=O bonds, which are linked to aromatic, olefinic, and functional structures typical of terpenoids, phenylpropanoids, and oxygenated compounds [5,31,32].

Raman bands identified in each essential oil are reported, aiming to associate the observed peaks with the functional groups and predominant metabolites of each species. In the essential oil of *T. filifolia*, vibrational modes were identified between 641 and 1656 cm^−1^. Signals in the 647–854 cm^−1^ range corresponded to out-of-plane C–H bending in cyclic rings, primarily associated with monoterpenes [24,33]. Vibrations between 1186 and 1318.5 cm^−1^ were assigned to C–H and C–O stretching, typical of esters, ketones, and aromatic phenols, and are linked to the presence of oxygenated phenylpropanoids [34]. Finally, bands in the 1609–1670 cm^−1^ range corresponded to C=C and C–H stretching, reflecting oxygenated aromatic compounds [21,27,35].

The essential oil of *A. citrodora* exhibited bands between 242 and 2916 cm^−1^. Signals below 750 cm^−1^ were associated with carbon skeleton deformations and out-of-plane C–H vibrations. C–O vibrations, typical of acyclic terpenic alcohols, were observed between 809 and 1154 cm^−1^, confirming the presence of oxygenated compounds in the oil [36,37]. Between 1310 and 1460 cm^−1^, C–O and CH_2_/CH_3_ deformations were observed, while bands at 1638–1680 cm^−1^ corresponded to C=C stretching of unsaturated compounds [34]. The high-intensity band at 2920–1930 cm^−1^ reflected C–H stretching vibrations associated with oxygenated monoterpenes, particularly alcohols or phenolic derivatives [36,37].

The essential oil of *E. globulus* showed peaks in the range of 550 to 2971 cm^−1^. Bands between 550 and 815 cm^−1^ were attributed to ring bending and out-of-plane C–H deformations characteristic of monocyclic terpenoids. Between 1170 and 1318 cm^−1^, C–H deformations of aromatic compounds were identified, whereas the band at 1457.05 cm^−1^ represented CH_2_/CH_3_ bending associated with aliphatic groups present in hydrocarbons and terpenoids. Finally, the signal at 2936.8 cm^−1^ corresponded to C-H stretching of unsaturated compounds such as alkenes and aromatic structures [23,33,38].

In the essential oil of *C. citratus*, bands between 1100 and 1340 cm^−1^ were related to symmetric CH_3_ deformations and C–C stretching (terpenic compounds), while bands at 1632–1681 cm^−1^ corresponded to C=C stretching of unsaturated aldehydes (citral isomers) [39,40]. The signal at 2928 cm^−1^ reflected C–H stretching, indicating the presence of conjugated aldehydes typical of monoterpenes or oxygenated phenylpropanoids, contributing to the citrus aroma and antioxidant capacity [41,42]. These signals are consistent with profiles reported for *Cymbopogon* oils with a citral-dominant profile [43].

In the essential oil of *C. nobile*, the band at 477.3 cm^−1^ was assigned to carbon skeleton bending modes, characteristic of cyclic and terpenic structures. Signals at 1021.7 and 1078.84 cm^−1^ corresponded to C–C stretching of the ring, indicating the presence of aromatic and cyclic rings typical of oxygenated monoterpenes and sesquiterpenes [44]. The C=C stretching at 1641.75 cm^−1^ indicated the presence of conjugated double bonds typical of aromatic sesquiterpenes. Finally, the peak at 2928.09 cm^−1^ corresponded to C-H stretching, consistent with oxygenated terpenoids [45].

The essential oil of *P. aduncum* exhibited bands between 262 and 770 cm^−1^, corresponding to carbon skeleton bending, and between 1294 and 1460 cm^−1^, associated with olefinic CH_3_ deformations. Between 1500 and 1600 cm^−1^, bending vibrations of CH_2_ and CH_3_ groups (δCH_2_/CH_3_) and C=C stretching were observed, corresponding to cyclic and unsaturated monoterpenes as well as oxygenated phenylpropanoids [45]. High-frequency bands at 1650.6, 2923.7, and 2987.4 cm^−1^ were attributed to C=C and aliphatic C–H stretching vibrations, which are characteristic of monoterpenes and sesquiterpenes. These signals, indicate the presence of oxygenated and unsaturated compounds typically found in this essential oil [46].

The essential oil of *M. mollis* was characterized by bands ranging from 303 to 2921 cm^−1^. Signals between 303 and 680 cm^−1^ were associated with ring deformations characteristic of α-pinene and limonene, while bands at 860 and 1065.65 cm^−1^ corresponded to out-of-plane C–H deformations and ν(C–C) + δ(C–H) stretching [47]. The band at 1309.72 cm^−1^ was assigned to C–O stretching, typical of alcohols, and the band at 1459.25 cm^−1^ was attributed to CH_2_ scissoring δ(C–H) [5]. Finally, signals at 2921 and 2940 cm^−1^ were assigned to aliphatic C–H stretching of cyclic monoterpenes, consistent with Raman profiles reported for essential oils of the genus *Mentha* [48].

The essential oil of *R. officinalis* exhibited peaks between 600 and 800 cm^−1^, attributed to ring deformations and out-of-plane C–H modes characteristic of cyclic monoterpenes such as α- and β-pinene, as well as 1,8-cineole [49,50]. Additionally, the signal at 1294.6 cm^−1^ was related to =CH wagging typical of olefinic bonds, while the band at 1378.9 cm^−1^ corresponded to CH_2_/CH_3_ bending associated with aliphatic chains of monoterpenes. The band at 1444.6 cm^−1^ was assigned to δ(C–H), also linked to the aliphatic skeleton. The band at 1638.7 cm^−1^ was attributed to C=C stretching, and the band at 2932.5 cm^−1^ to C–H stretching (double bond), both indicative of unsaturated structures characteristic of monoterpenes and sesquiterpenes present in the essential oil [50,51,52].

### 2.2. Multivariate Analysis of Raman Spectra

Principal component analysis (PCA) applied to the Raman spectra allowed a clear differentiation of the analyzed essential oils, showing a distinct separation between species (Figure 2a). The first two principal components, PC1 and PC2, collectively explained 75.73% of the total variance (40.25% and 35.48%, respectively), demonstrating the model’s adequate capacity to represent chemical differences between species.

The essential oils of *E. globulus* and *R. officinalis* formed closely related clusters, reflecting their structural similarity associated due to presence of oxygenated monoterpenes [23,50]. Similarly, *M. mollis* and *P. aduncum*, also clustered together, which is consistent with our results, as both share oxygenated functional groups (alcohols, ketones, and esters) and have a composition rich in monoterpenes and sesquiterpenes [53]. Another cluster, composed of *C. citratus* and *A. citrodora*, was identified, suggesting a high structural similarity between these oils, although distinct from the previously mentioned groups due to their greater separation in the projection plane. Finally, although *T. filifolia* was positioned close to *M. mollis* due to similarity in ester and alcohol content, the *M. mollis* oil exhibited a more dispersed distribution, likely associated with its complex composition of phenylpropanoids, unsaturated hydrocarbons, and terpenic rings [1,54], resulting in a more heterogeneous spectrum.

The Hotelling T^2^ versus residual Q plot (Figure 2b) allowed verification of the PCA model validity (95% confidence limit). The observed distribution shows that most species are located within the optimal fit region (lower left quadrant), indicating that the model adequately describes the variability of their chemical characteristics. The dispersion of *C. nobile* and *T. filifolia* in the lower right quadrant highlights chemical compositions that distinguish them from the rest. The absence of outliers in the critical region confirms that there are no observations with high influence (elevated T^2^) and high fitting error (high residual Q), supporting the stability and robustness of the PCA model.

Hierarchical cluster analysis (Figure 2c) complemented the PCA results by showing the similarity relationships between species through hierarchical clustering based on weighted variance distance. The dendrogram revealed the formation of two main groups. The first group consisted of *A. citrodora* and *C. nobile*, characterized by a predominance of oxygenated monoterpenes [55]. The second group included *E. globulus*, *P. aduncum*, *R. officinalis*, *M. mollis*, *T. filifolia*, and *C. citratus*, whose Raman signals were dominated by aromatic compounds and aliphatic C–H stretching [18,28,56].

This hierarchical organization was consistent with the chemical structure of the oils, as it clustered species with a high proportion of oxygenated monoterpenes together, while separating those dominated by aromatic or aliphatic compounds. Overall, the agreement between the results obtained from PCA, T^2^–Q, and HCA confirms that Raman spectroscopy, combined with multivariate analysis, constitutes an effective tool for discriminating essential oils based on their molecular composition and botanical origin.

### 2.3. Gas Chromatography–Mass Spectrometry (GC–MS) Results Analysis

After characterizing the essential oils (EO) using Raman spectroscopy, which allowed the identification of their main functional groups and vibrational profiles, chemical analysis was conducted by gas chromatography coupled with mass spectrometry (GC–MS). After characterizing the essential oils (EO) using Raman spectroscopy, which allowed the identification of their main functional groups and vibrational profiles, chemical analysis was conducted by gas chromatography coupled with mass spectrometry (GC–MS). This technique enables the precise identification, and a relative quantification of volatile compounds present in the EO, complementing the information obtained from Raman spectroscopy by providing data on the specific molecular composition of chemical families.

It is important to note that the GC-MS data correspond to semi-quantitative relative peak areas (%), obtained after signal normalization. Therefore, the reported values represent the relative abundance of each compound in the oil and should not be interpreted as absolute concentrations or gravimetric percentages. This method allows for a reliable comparison of the relative chemical composition between essential oils, although it does not consider the response factors of each detector.

GC–MS analysis identified a total of 224 compounds (Appendix A) belonging to nine chemical families: alcohols, aldehydes, esters, hydrocarbons, ketones, phenols, phenylpropanoids, terpenoids, and a residual group classified as “others,” in the EO of *T. filifolia*, *A. citrodora*, *E. globulus*, *C. citratus*, *P. aduncum*, *M. mollis*, and *R. officinalis*. In most essential oils, terpenoids constituted the predominant fraction, representing over 70% of the total relative peak area (Figure 3), followed by lower proportions of alcohols, hydrocarbons, ketones, and esters. This pattern aligns with the findings of Fenghour & Bouabida [25], who report terpenoids as the main constituents of EO, with abundant compounds such as α-curcumene (7.84–16.91%) in *R. officinalis* [10], β-cyclocitral (21.46% relative peak area), germacrene D (11.7–62.9% relative peak area), and β-caryophyllene (17.34% relative peak area) in *P. aduncum* [57], as well as α-terpinyl acetate and citral (17% relative peak area) in *A. citrodora* [58].

The essential oil of *R. officinalis* exhibited notable relative peak areas of D-limonene (8.59%), linalool (2.91%), α-terpinene (2.51%), and β-caryophyllene (18.26%) (Table 1). These values are higher than those reported by Alvarado-García et al. [53], who identified 10.76% relative peak area of β-caryophyllene in *R. officinalis* samples through GC–MS analysis. In contrast, Malik & Upadhyay [59] recorded only 1.54% relative peak area of this compound in EO from *R. officinalis* originating from India. Similarly, the essential oil of *P. aduncum* presented high semiquantitative contributions based on GC–MS signal intensity of β-caryophyllene (11%), D-limonene (4.59%), and β-pinene (3.37%), slightly higher than those reported by Jedidi et al. [60] in samples collected in northwestern Tunisia, where a β-caryophyllene relative peak area of 9.49% was reported.

The observed variations in β-caryophyllene relative peak area can be attributed to both environmental factors and post-harvest and processing conditions. On one hand, parameters such as temperature, humidity, altitude, and soil composition in the cultivation areas directly influence the synthesis and accumulation of terpenoids [10]. On the other hand, technological aspects related to drying and extraction methods also modify the chemical profile of the oil. In this regard, Mohammed et al. [61] demonstrated that prolonged drying of *R. officinalis* reduces the semiquantitative contribution of volatile compounds, mainly monoterpenes such as D-limonene, linalool, and α-terpinene, while also altering the proportion of sesquiterpenes such as β-caryophyllene. Similarly, Zhang et al. [62] showed that the extraction method (hydrodistillation, steam distillation, or solvent extraction) significantly modifies the EO relative peak areas, whereas Zheljazkov et al. [63] reported that both the distillation time and the sample condition (dry or fresh) affect the yield and bioactivity of *R. officinalis* essential oil.

Additionally, the part of the plant used for essential oil extraction is also a determining factor. In *R. officinalis*, it has been observed that young leaves concentrate higher levels of monoterpenes, whereas structural tissues such as stems and bark present lower proportions of these metabolites [64]. Similarly, in species of the genus *Origanum*, EO extracted from leaves show higher concentrations of volatile aromatic compounds, such as carvacrol (30.73%), thymol (18.81%), and β-caryophyllene (8.21%), compared to those obtained from stems, where the concentrations were lower (carvacrol 6.02%, thymol 3.46%, and β-caryophyllene, which was not detected) [65].

Likewise, in *P. aduncum*, the plant part also influences the semiquantitative profile of the essential oil. For instance, Valadares et al. [66] compared essential oils obtained from leaves and flowers, finding differences in the relative peak areas of β-caryophyllene (7.2% in flowers and 5.4% in leaves) and other minor compounds such as terpinen-4-ol and myrcene. Moreover, antifungal activity varied depending on the plant organ, as the oil obtained from flowers completely inhibited the growth of *Sclerotinia sclerotiorum* at doses above 30 µL, whereas the leaf essential oil required 50 µL to achieve 98.74% inhibition. These findings confirm that the distribution of monoterpenes and sesquiterpenes is closely associated with the structural characteristics of different plant organs, highlighting the importance of considering the plant part used in EO characterization.

The essential oil of *M. mollis* exhibited a unique composition, characterized by the dominant presence of cis,cis-nepetalactone at concentrations above 20%, a compound absent in the other evaluated EO in this study. The absence of β-caryophyllene, along with the predominance of this lactone, highlights the metabolic uniqueness of *M. mollis* oil compared to other members of the Lamiaceae family. This finding represents a significant chemical discovery, as this study constitutes the first report of cis,cis-nepetalactone in this species; previous investigations have not documented the presence of this lactone in *M. mollis* [1,54,67,68].The detection of this lactone in *M. mollis* essential oil differentiates the chemical composition of the plant, which is relevant from both a chemical and functional perspective. This finding suggests the existence of a population with distinct chemical characteristics, whose profile could be less toxic and potentially more bioactive than other *M. mollis* varieties dominated by pulegone [69,70].

Alongside nepetalactone, linalool (9.63%) and D-limonene (6.87%) were identified as relevant secondary components, with relative peak areas higher than those reported by Cravero Ponso et al. [1] for the same species (<5%). These oxygenated monoterpenes are known for their antibacterial, anti-inflammatory, anxiolytic, and contraceptive properties, reinforcing the bioactive potential of the essential oil. The differences in EO relative peak areas compared to previous studies are mainly attributed to natural variations in plant chemistry and the environmental conditions in which the plants grew, such as climate, soil type, and altitude. For example, van Baren et al. [54] reported marked variations in linalool relative peak areas (6–84%) among Argentine populations of *M. mollis*, while Quispe-Sanchez et al. [17] documented higher proportions of D-limonene (14.57%) and linalool (15.30%) in samples collected in the same Andean region as this study (Peru), suggesting that even under similar environmental conditions, significant differences can arise due to genetic, phenological, or methodological factors, including the collection period, which is associated with the plant’s developmental stage and seasonal environmental conditions.

Comparatively, the composition observed in this study differs from the classical profiles of *M. mollis*, where pulegone, menthone, and 1,8-cineole are the characteristic components [71,72,73,74]. The analysis of the essential oil of *C. citratus* allowed the identification of linalool (2.33%) and geraniol (6.06%) relative peak areas. Although the overall profile of the species is characterized by the predominance of citrals (a mixture of geranial and neral), the detection of these terpenoids at proportions above 1% is relevant due to their contribution to both the aroma and the bioactive properties of the oil essential. For instance, studies indicate that the presence of geraniol in *C. citratus* can enhance the antibacterial activity of the essential oil [1,15,54]. Monoterpenic alcohols have been shown to disrupt bacterial cell membrane integrity, promoting ion leakage and disorganization of enzymatic systems, thereby increasing efficacy against Gram-positive bacteria (*Staphylococcus aureus* and *Bacillus subtilis*) and Gram-negative bacteria (*Escherichia coli*, *Pasteurella multocida*) [28,75,76]. The absence or low relative peak areas of other chemical compounds (β-caryophyllene, myrcene, citral) in the essential oil may favor the stability of active components such as geraniol and linalool, preserving their antibacterial activity and preventing potential interactions that could reduce efficacy [77].

The essential oil of white *E. globulus* exhibited a chemical profile dominated by myrcene (8.49%), β-pinene (8.21%), and terpinen-4-ol (3.54%), relative peak areas higher than those reported in other species of the genus. For instance, Belhachemi et al. [78] reported the presence of these terpenoids in red *E. globulus* essential oil, albeit at lower relative peak areas (<2%). Similarly, dos Santos et al. [79] reported low β-pinene levels in lemon-scented *E. globulus* (1.98%), values much lower than those observed in the present study. Additionally, Ait benlabchir et al. [23] detected myrcene (1.51%) and terpinen-4-ol (2.43% in directly distilled oil and up to 15.24% in purified fractions) in white *E. globulus* essential oils, demonstrating that although these compounds are usually present at low to moderate relative peak areas, their values can vary widely depending on the fraction analyzed or the processing method applied.

In Mediterranean regions, a comparative analysis of *E. globulus* essential oil showed that myrcene and β-pinene rarely exceed 2–3%, while terpinen-4-ol remains below 5% [26]. Likewise, studies on commercial *E. globulus* oils indicate that these monoterpenes are secondary components: myrcene (<2%), β-pinene (1–2%), and terpinen-4-ol (1%) [80]. In contrast, the relative peak areas of these compounds in this study are significantly higher.

The high proportion of β-pinene and myrcene observed in the study could be explained by the presence of specific chemotypes of *E. globulus*, such as those characterized by the combinations γ-terpinene + o-cymene + β-pinene or α-pinene + 1,8-cineole. These chemotypes correspond to populations that produce essential oils with differentiated chemical profiles, resulting from genetic or environmental variations specific to each group [81]. Additionally, it is possible that the metabolic pathways responsible for monoterpene formation favored the production of β-pinene and myrcene in this population over other compounds. In this regard, it has been demonstrated in similar species, such as tea tree (*Melaleuca alternifolia*), that the activity of genes associated with the methylerythritol phosphate (MEP) pathway, which generates the precursors necessary for monoterpene synthesis, directly influences both the quantity and type of monoterpenes produced, supporting this possible explanation [82].

The essential oils of *A. citrodora* and *T. filifolia*, both recognized for their high content of biologically active monoterpenes, exhibited differentiated chemical compositions. In the essential oil of *A. citrodora*, the predominant compounds were citronellol (10.92%), β-caryophyllene (6.0%), and D-limonene (6.31%), whereas in *T. filifolia*, citral (2.35%), linalool (1.61%), and D-limonene (2.81%) were the main components. In the case of *A. citrodora*, β-caryophyllene and D-limonene were present at similar relative peak areas; however, D-limonene showed notable differences between the two oils, with a higher proportion in *A. citrodora* compared to *T. filifolia*.

In contrast with our findings, Al-Maharik et al. [10] reported for the essential oil of *A. citrodora* the presence of β-caryophyllene (0.62%), citral (10.79%), limonene (0.21%), and D-limonene (19.03%), along with other minor compounds such as 1-terpineol (0.07%), 4,4-dimethyl-2-butenolide (0.17%), α-thujone (0.16%), and ethyl 3-(2-furyl)propanoate (0.09%). Although the qualitative composition (the same compounds) partially coincides with that of our study, the quantitative differences in relative peak areas could be explained by the effects of cultivation factors such as temperature and light intensity, which modulate the expression of genes encoding terpene synthases enzymes that determine both the biosynthetic pathway and the proportion in which precursors are converted into different monoterpenes [83,84].

For instance, it has been demonstrated that the expression of monoterpene biosynthetic genes can vary depending on the time of day, increasing around midday when light and temperature are higher, coinciding with peaks in the production of terpenoids such as D-limonene or linalool in lavender [23]. Similarly, studies in other species, including alpine plants [85] and sandalwood [86], have observed that abrupt changes in temperature or radiation levels affect volatile terpene emission, as lighter or less-oxygenated compounds can be lost or rearranged in response to thermal stress. Therefore, it is likely that in the analyzed population of *A. citrodora*, local conditions—such as high daytime temperatures or intense irradiance—favored the biosynthetic preference toward monoterpenes like D-limonene or citral, resulting in higher relative peak areas than those reported by Al-Maharik et al. [10].

Similarly, Rashid et al. [30] reported in the same species levels of β-caryophyllene (5.09%), D-limonene (18.80%), and linalool (0.38%). In this case, the relative peak area of β-caryophyllene was comparable to that observed in our study, suggesting that this compound exhibits low variation throughout the plant’s vegetative cycle; however, this assumption remains to be verified. In contrast, the variations observed in D-limonene and linalool could be explained by differences in the availability of precursor compounds in the plant and by changes induced by temperature and solar radiation. Recent studies have demonstrated that environmental and physiological conditions directly modify the proportion of these compounds [87]. For example, in *Zanthoxylum bungeanum*, D-limonene content increases under higher temperature and solar radiation, whereas linalool decreases under the same conditions, indicating an inverse relationship between these two compounds [88,89].

Moreover, Dumitrescu et al. [90] analyzed the essential oil of *T. filifolia*, identifying linalool (1.90%) and D-limonene (10.01%) as the main components. Compared to our results, the relative peak area of linalool was similar, whereas D-limonene was considerably higher. Beyond extraction methodological differences, this variation can be attributed to several physiological factors, such as leaf characteristics and soil properties, which directly influence the relative peak area of the essential oil. In particular, the physiological maturity and position of leaves on the stem at harvest significantly affect the proportion of monoterpenes. In plants such as geranium, younger leaves exhibit higher biosynthetic activity and accumulate greater levels of alcohols like linalool, whereas older leaves tend to present a more hydrocarbon-rich profile, with predominance of limonene [91]. Additionally, soil properties, especially pH and mineral composition, modify the availability of essential nutrients and the efficiency of enzymes involved in monoterpene synthesis [92].

Unlike the other essential oils, the essential oil of *C. nobile* could not be characterized by GC–MS due to its low extraction yield and the high viscosity of the resulting concentrate (Appendix A), which exhibited a pasty and dense consistency, that prevented adequate volatilization during chromatographic analysis. This behavior can be attributed to the predominance of higher molecular weight and less volatile compounds, such as oxygenated sesquiterpenes, high-molecular-weight esters, or traces of natural waxes [93,94], which hinder their detection and separation by gas chromatography.

### 2.4. Multivariate Analysis of GC-MS Data

Multivariate analysis using Principal Component Analysis (PCA) and Hierarchical Cluster Analysis (HCA) allowed the identification of chemical relationships among the essential oils evaluated and the determination of the functional groups responsible for their differentiation. Overall, the first two dimensions of the PCA (Figure 4) explained 72.7% of the total variance (PC1 = 53.1% and PC2 = 19.6%), demonstrating an adequate representation of the chemical variability of the samples.

The essential oil of *T. filifolia* was projected at the far right, close to the vectors of aldehydes and hydrocarbons, indicating a composition rich in these compounds and a clear differentiation from the other oils. In contrast, *M. mollis* was located at the upper part of the plot, associated with the vectors of alcohols and esters, suggesting a composition dominated by more polar oxygenated metabolites [95,96]. The oils of *E. globulus* and *R. officinalis* clustered in the left region, linked to terpenoids and phenolic compounds, consistent with their characteristic content of oxygenated monoterpenes, in agreement with Čmiková et al. [97], who highlight 1,8-cineole as a main component. *P. aduncum* was projected in the lower right quadrant, defined by phenylpropanoids, while *C. citratus* was positioned toward the lower left quadrant, in the area associated with the “others” group, suggesting a distinctive chemical profile with minor or species-specific compounds.

These groupings observed in the PCA were confirmed by hierarchical cluster analysis (Figure 4c), which revealed three main levels of chemical similarity. The oil of *T. filifolia* exhibited a singular profile dominated by aldehydes and hydrocarbons, based on relative peak areas (%). Secondly, *P. aduncum* formed an independent group, consistent with its predominance of phenylpropanoids [98]. Finally, the cluster formed by *M. mollis*, *E. globulus*, *C. citratus*, *A. citrodora*, and *R. officinalis* showed the highest similarity, reflecting a more homogeneous composition based on terpenoids and alcohols, as reported in this study.

The multivariate analysis demonstrated that the differences in the distribution of essential oils are driven by the chemical nature of their dominant metabolites. Thus, PCA and cluster analysis not only confirmed the consistency among the reported chemical compositions but also facilitated the identification of characteristic patterns that distinguish each species based on its profile of terpenoids, phenylpropanoids, aldehydes, or alcohols.

During the spectroscopic analysis, additional bands within the 200–3100 cm^−1^ range were also identified, associated with families such as aldehydes, esters, alcohols, and organic acids. Complementarily, the GC–MS analysis allowed the detection of compounds from diverse families, including ketones and phenylpropanoids, in addition to the major terpenoids. However, these metabolites were not discussed in detail in the present study because their relative concentrations were considerably lower, and the main focus was on the characterization and interpretation of the major metabolites (primarily terpenoids), as they largely determine the chemical profile, bioactive properties, and variability among species.

It should be noted that the hierarchical classification obtained from Raman spectroscopy (Figure 2c) differs from the GC–MS data (Figure 4c). This is understandable, given that Raman detects vibrational characteristics related to functional groups and molecular structures, while GC–MS provides semi-quantitative information on specific volatile compounds [99]. Therefore, the classification patterns obtained with each technique should be interpreted in a complementary manner, rather than combined directly.

### 2.5. Limitations and Future Perspectives

The complementary analysis of Raman spectroscopy and gas chromatography–mass spectrometry (GC–MS) enabled the identification of the main metabolites in the EO of *P. anisum*, *A. citrodora*, *E. globulus*, *C. citratus*, *M. chamomilla*, *P. aduncum*, *M. mollis*, and *R. officinalis*. Terpenoids were detected that clearly differentiate each species. In *P. anisum*, citrals (2.35%) and D-limonene (2.81%) were observed. In *A. citrodora*, citronellol (10.92%) and β-caryophyllene (6%) were predominant, whereas *E. globulus* was characterized by myrcene (8.49%) and β-pinene (8.21%). *C. citratus* showed geraniol (6%), and *P. aduncum* was distinguished by β-caryophyllene (11%). *M. mollis* was notable for a high semiquantitative contribution based on GC–MS signal intensity of cis,cis-nepetalactone (30.16%), while *R. officinalis* exhibited a predominance of β-caryophyllene (18.26%).

The high content of these terpenoids suggests the anti-inflammatory, antioxidant, sedative, muscle-relaxant, antimicrobial, and antitumor properties attributed to these plants [78,100]. For example, β-caryophyllene has been described as an insecticidal agent, a bacterial biofilm inhibitor, and a modulator of apoptosis in cancer cells [101,102]. Additionally, da Costa Sobral et al. [103] demonstrated that the combination of β-caryophyllene with drugs such as pregabalin enhances seizure control, whereas Nunes et al. [104] reported that camu camu essential oil, rich in (E)-caryophyllene (14.02–34.31%), exhibits anticancer and anti-inflammatory potential by inhibiting tumor cells and reducing inflammatory processes in vitro.

The discovery of cis,cis-nepetalactone in the essential oil of *M. mollis* represents the first record of this lactone in the species. Its presence indicates the existence of populations with distinct chemical profiles, likely influenced by genetic or environmental factors affecting the production of bioactive compounds [77,105]. This finding opens new research avenues to evaluate how genetics and environmental conditions influence the production of cis,cis-nepetalactone and other bioactive compounds in *M. mollis.* Understanding these factors could allow the identification of previously unstudied varieties with higher semiquantitative contributions based on GC–MS signal intensity of functional metabolites and low toxicity risk. Additionally, these essential oils could have applications in natural cosmetics and aromatherapy, particularly due to their antimicrobial, anti-inflammatory properties and characteristic fresh aroma, further highlighting their functional and commercial potential [67,106].

Given that reports of nepetalactone in Lamiaceae family are scarce, it is recommended to confirm its structural identification through retention index comparison, co-injection with authentic standards, and the use of complementary techniques such as GC-MS/MS or NMR [107,108,109]. This approach will validate the authenticity of the compound and consolidate its role as a chemical marker distinguishing *M. mollis* in Peru.

Although the techniques employed enabled the characterization of the essential oils, Raman spectroscopy showed limitations in differentiating structural isomers or stereoisomers and could not detect compounds or chemical families present at very low concentrations. Likewise, GC–MS encountered difficulties with dense or low-volatility oils, such as that of *C. nobile*, restricting the complete identification of all constituents. Future studies are therefore recommended to optimize analytical methods for detecting minor or low-volatility compounds not captured by current techniques; to investigate how environmental conditions in different growing areas may influence the biosynthesis of volatile compounds—potentially leading to the formation of distinct ecotypes; and to evaluate the industrial applications of these essential oils in light of their unique chemical composition and bioactive properties.

## 3. Materials and Methods

### 3.1. Plant Material and Sampling

Eight species of aromatic medicinal plants were studied, as detailed in Table 2. The analyzed species were anise (*T. filifolia*), lemon verbena (*A. citrodora*), eucalyptus (*E. globulus*), lemongrass (*C. citratus*), matico (*P. aduncum*), poleo (*M. mollis*), and rosemary (*R. officinalis*) (Figure 5 and Appendix A). The taxonomic identification of the species was verified by the Kuelap Herbarium of the Universidad Nacional Toribio Rodríguez de Mendoza (UNTRM). Plant material was collected during the peak flowering stage, obtaining stems, leaves, and flowers depending on the species, between February and March 2024, in the Amazonas region, Peru. The species *T. filifolia*, *E. globulus*, *C. citratus*, *M. mollis*, *C. nobile*, *P. aduncum*, and *R. officinalis* were collected directly from the field, whereas *A. citrodora* was purchased fresh from the central market of Chachapoyas. For each species, 15 kg of plant material were collected, distributed in three replicates of 5 kg each. The collected leaves were transported in containers to the Engineering Laboratory of UNTRM for washing and subsequent air-drying at 21 °C, with an average drying time of two days, prior to distillation.

### 3.2. Essential Oil Extraction (EO)

The extraction of essential oils (EO) from the eight aromatic medicinal plants was performed following the procedure described by Vasquez-Gomez et al. [110], with minor modifications. Steam distillation was carried out using a TECNAL TE-2761 apparatus (Piracicaba, Brazil) at a temperature of 130 °C. For each species, 15 kg of plant material were processed in three independent replicates, with a distillation time of 4 h per sample. The water–oil mixture obtained after distillation was cooled to 1 °C to promote phase separation (aqueous and oily layers). The essential oil phase was then carefully decanted and collected for each species. The EO were stored in amber glass vials with airtight caps at 4 °C until analysis.

The essential oil yield (Appendix A) was calculated according to Barbosa et al. [57] using the following equation(1)$$Y\%=\frac{V}{W}*100$$
where *Y* is the essential oil yield (%, *v*/*w*), *V* is the volume of essential oil obtained (mL), and *W* is the weight of the plant material used (g).

### 3.3. Raman Spectral Data Acquisition

Raman spectra of the essential oils from the eight aromatic medicinal plants were acquired using a Horiba XploRA Plus confocal Raman microscope system (Horiba Scientific, Montpellier, France), equipped with an Olympus BX41 microscope (Olympus, Tokyo, Japan) and a 50× long working distance (LWD) objective. Excitation was performed with a 532 nm laser operating at 40 mW power, using a 1200 lines/mm diffraction grating. The Raman signal was detected with a charge-coupled device (CCD) detector, covering an extended spectral range from 100 to 5000 cm^−1^ with a spectral resolution of 3 cm^−1^ [111].

For each measurement, approximately 20 µL of essential oil was placed in an aluminum crucible. Spectra were recorded with an integration time of 5 s and 10 spectral accumulations, yielding a total of 30 spectra per sample. To minimize sample heterogeneity, three independent replicates were analyzed for each essential oil, resulting in a total of 90 spectra per species. Raman peaks were assigned according to previously published literature.

### 3.4. Chemical Characterization by Gas Chromatography–Mass Spectrometry (GC–MS)

The chemical composition of the essential oils (EO) was determined using gas chromatography coupled with mass spectrometry (GC–MS) (GC System 7890B, MSD 5977B, Agilent Technologies, Santa Clara, CA, USA) following Adams [112]. The EO were diluted (1 μL of essential oil + 49 μL of hexane), and 0.5 μL of the resulting solution was injected in splitless mode. Chromatographic separation was performed on a DB-5MS UI capillary column (60 m × 0.25 mm × 1.0 μm). Helium was used as the carrier gas at a constant flow rate of 1 mL/min. The injector, detector, transfer line, and ion source were maintained at 220, 150, 240, and 280 °C, respectively. The oven temperature was set at 60 °C, then increased at a rate of 3 °C/min up to 246 °C and held for 15 min. Mass spectra were recorded in scan mode over a mass range of 40–600 *m*/*z*. Detected compounds were identified by comparison with the National Institute of Standards and Technology Library (NIST Library 17). Data were deconvoluted using MassHunter Unknowns Analysis software (version 10.1 build 10.1.733.0, Agilent Technologies), considering a similarity index ≥ 80%. The linear retention index (LRI) was calculated to confirm compound identification by injecting a commercial n-alkane standard mixture containing (C8–C20, ~40 mg/L each, in hexane, Supelco, Sigma-Aldrich, St. Louis, MO, USA) under the same chromatographic conditions applied to the essential oil samples.

The data obtained by GC-MS were processed by integrating the areas of the peaks of the total ion chromatogram (TIC) corresponding to each compound. The areas were normalized with respect to the total signal and expressed as relative peak areas (%), which provided semi-quantitative information that was subsequently used in multivariate analyses (PCA and HCA).

### 3.5. Analysis of Raman and GC–MS Spectral Data

Raman spectral data from the essential oils of the eight aromatic medicinal plants were preprocessed following the protocols described by Castro-Alayo et al. [113]. The preprocessing included data smoothing, baseline correction (200–3100 cm^−1^), vector normalization, and substrate removal using Solo+MIA 8.1 software (Eigenvector Research, Inc., Wenatchee, WA, USA).

Subsequently, a principal component analysis (PCA) was performed to reduce data dimensionality and visualize clustering patterns among the essential oils based on their chemical composition. Hotelling’s T^2^ and Q residual statistics were computed at a 95% confidence level to identify and remove anomalous spectra considered as outliers. Additionally, a hierarchical cluster analysis (HCA) was conducted to complement the PCA and confirm the formation of distinct groups among the essential oils. The combination of these multivariate methods enabled the identification of similarity patterns and validation of the obtained groupings.

For the multivariate processing of GC–MS data, the detected compounds were classified according to their chemical families, and PCA and HCA were again applied using RStudio software (version 4.3.3, Boston, MA, USA). This approach facilitated the identification of similarity patterns, visualization of clustering based on chemical composition, and verification of group consistency, thereby complementing the results obtained from Raman analysis.

## 4. Conclusions

The integration of Raman spectroscopic and chromatographic (GC–MS) analyses enabled a comprehensive characterization of essential oils obtained from eight aromatic medicinal species (*T. filifolia*, *A. citrodora*, *C. citratus*, *E. globulus*, *C. nobile*, *P. aduncum*, *M. mollis*, and *R. officinalis*). The results revealed that interspecific differences were primarily associated with the relative proportion of oxygenated compounds (alcohols, aldehydes, and esters) versus aromatic hydrocarbons and terpenoids. Raman spectra exhibited characteristic bands corresponding to C–H, C=C, C–O, and C=O stretching vibrations, reflecting the presence of aromatic and olefinic structures typical of mono- and sesquiterpenes. These findings were corroborated by GC–MS, which identified 224 compounds across nine chemical families, with terpenoids predominating (>70% in most species). Major metabolites included β-caryophyllene (18.26% and 11% in *R. officinalis* and *P. aduncum*, respectively), citronellol (10.92% in *A. citrodora*), citral (2.35% in *T. filifolia*), geraniol (6.06% in *C. citratus*), β-pinene and myrcene (8.21% and 8.49% in *E. globulus*), and cis,cis-nepetalactone (30.16% in *M. mollis*). The detection of cis,cis-nepetalactone in *M. mollis* constitutes a significant scientific contribution, representing the first report of this lactone in the species and opening new perspectives for research on its structure and biological potential. Furthermore, the observed correlation between vibrational profiles and GC–MS quantifications highlights the effectiveness of the complementary Raman and GC–MS approach as a rapid, reproducible, and non-destructive method for essential oil characterization.

## Figures and Tables

**Figure 1 ijms-26-11254-f001:**
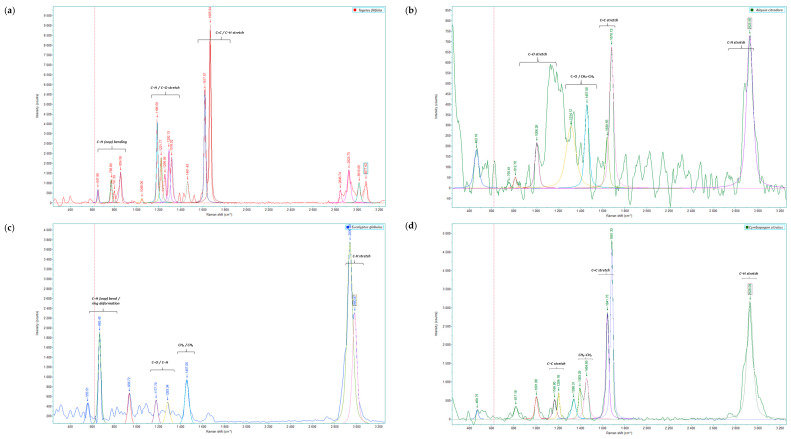
Representative Raman spectra of (**a**) *Tagetes filifolia*, (**b**) *Aloysia citrodora*, (**c**) *Eucalyptus globulus*, (**d**) *Cymbopogon citratus*, (**e**) *Chamaemelum nobile*, (**f**) *Piper aduncum*, (**g**) *Minthostachys mollis*, and (**h**) *Rosmarinus officinalis* in the spectral region of 200–3100 cm^−1^. The average spectra for each essential oil are presented in Appendix A.

**Figure 2 ijms-26-11254-f002:**
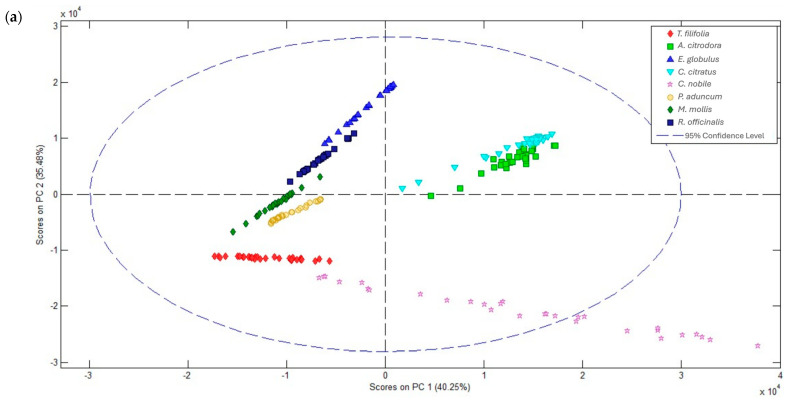
Principal Component Analysis (PCA) and Hierarchical Classification of Plant Samples: (**a**) PCA scores, (**b**) Q vs Hotelling’s T^2^ in the PCA model, and (**c**) dendrogram.

**Figure 3 ijms-26-11254-f003:**
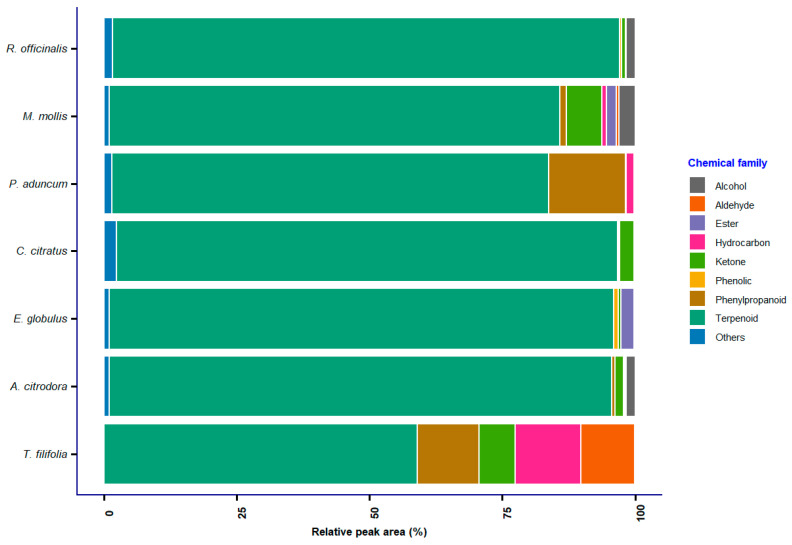
Distribution of chemical families identified in essential oils by GC–MS analysis.

**Figure 4 ijms-26-11254-f004:**
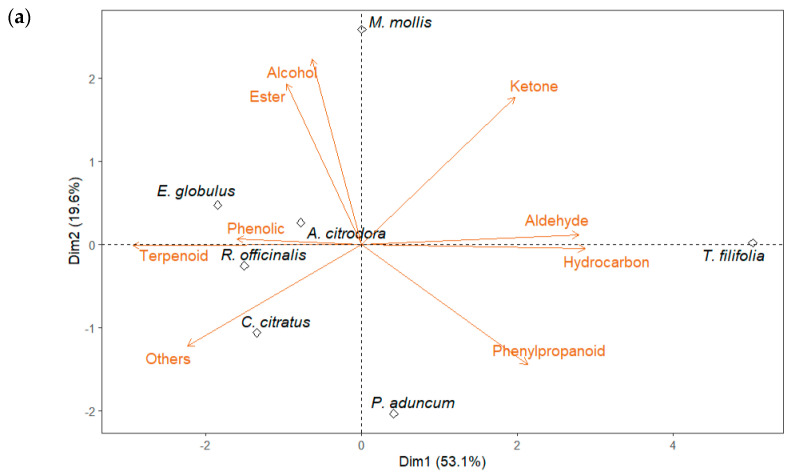
Principal Component Analysis: (**a**) biplot of individuals and explanatory variables, and (**b**) correlation circle, and Hierarchical Analysis: (**c**) dendrogram.

**Figure 5 ijms-26-11254-f005:**
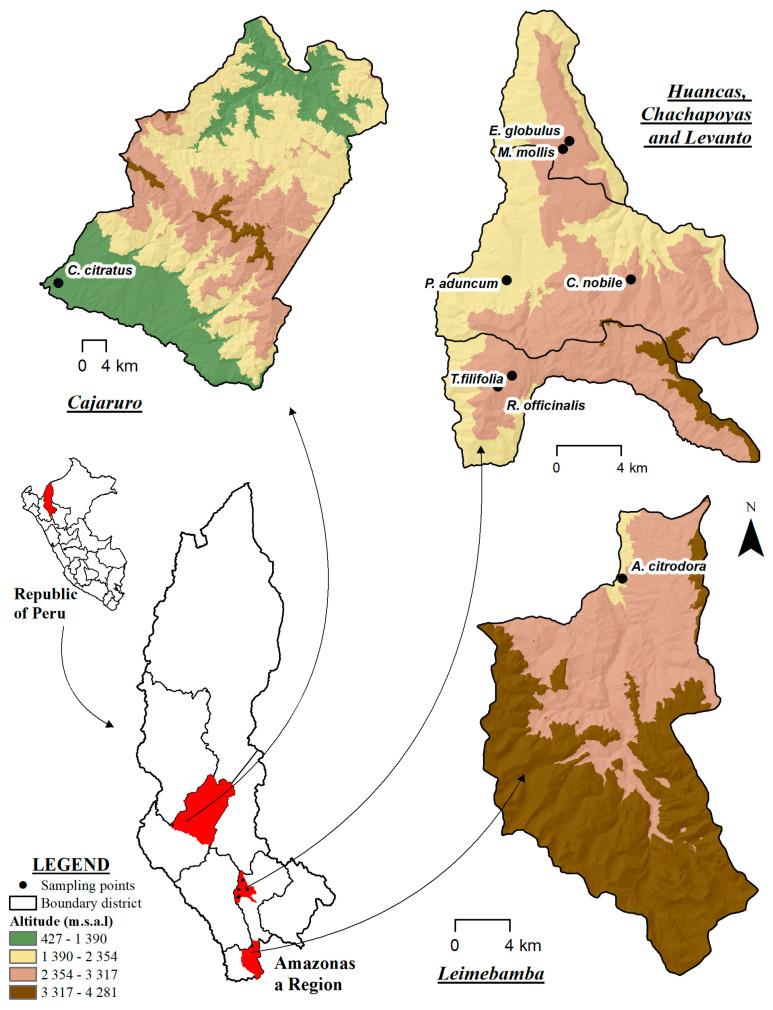
Growth location of aromatic plants.

**Table 1 ijms-26-11254-t001:** Areas of relative peaks (%) of the main terpenoids identified in the essential oils of seven medicinal species.

Group	Compound	*R. officinalis*	*M. mollis*	*P. aduncum*	*C. citratus*	*E. globulus*	*A. citrodora*	*T. filifolia*
Terpenoid	cis,cis-Nepetalactone	0.00	30.16	0.00	0.00	0.00	0.00	0.00
	α-terpinene	2.51	0.00	1.35	0.00	0.13	0.00	0.00
	Citral	0.00	0.00	0.00	0.00	0.40	0.00	2.35
	Myrcene	0.00	0.39	0.00	0.00	8.49	0.00	0.58
	Sabinene	0.49	0.86	0.00	0.00	0.00	0.00	0.00
	β-caryophyllene	18.26	0.00	11.00	0.22	0.00	6.00	0.00
	Citronellol	0.00	0.00	0.00	0.00	0.00	10.92	0.00
	Linalool	2.91	9.63	0.33	2.33	0.90	0.59	1.61
	Terpinen-4-ol	0.00	0.00	2.03	0.00	3.54	0.15	0.00
	Thymol	0.00	2.04	0.00	0.00	0.00	0.00	0.00
	D-limonene	8.59	6.87	4.59	0.28	0.00	6.31	2.81
	Eucalyptol	0.00	0.29	0.00	0.00	0.00	1.42	0.35
	Geraniol	0.00	0.00	0.00	6.06	3.13	3.17	0.39
	β-pinene	0.00	0.00	3.37	0.00	8.21	0.00	0.91

The table shows the relative peak areas (%) of the main terpenoids detected in essential oils by GC–MS, derived from signal normalization. The values represent semi-quantitative data.

**Table 2 ijms-26-11254-t002:** Taxonomic and geographical characterization of aromatic medicinal plants.

Common Name	Scientific Name	Abbreviation	Botanical Family	Growth Habit	Geographical Origin	Latitude–Longitude
Anise	*Tagetes filifolia*	*T.filifolia*	Asteraceae	Herb	Chachapoyas-Levanto	−6.30625, −77.8987
Lemon verbena	*Aloysia citrodora*	*A. citrodora*	Verbenaceae		Chachapoyas-Leymebamba	−6.7023, −77.7966
Eucalyptus	*Eucalyptus globulus*	*E. globulus*	Myrtaceae	Shrub	Chachapoyas-Huancas	−6.17500, −77.7966
Lemongrass	*Cymbopogon citratus*	*C. citratus*	Poaceae	Herb	Utcubamba-Bagua Grande	−5.7329, −78.4279
Matico	*Piper aduncum*	*P. aduncum*	Piperaceae	Shrub	Chachapoyas-Chachapoyas	−6.2528, −77.9016
Chamomile	*chamaemelum nobile*	*C. nobile*	Asteraceae	Herb		−6.2523, −77.8325
Pennyroyal	*Minthostachys mollis*	*M. mollis*	Lamiaceae	Shrub		−6.1795, −77.8702
Rosemary	*Rosmarinus officinalis*	*R. officinalis*	Lamiaceae	Herb	Chachapoyas-Levanto	−6.3121, −77.9066

## Data Availability

The raw data supporting the conclusions of this article will be made available by the authors on request.

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
