# Peer review of "Characterization of Terpenoids in Aromatic Plants Using Raman Spectroscopy and Gas Chromatography–Mass Spectrometry (GC–MS)"

_ijms, 2025, doi:10.3390/ijms262311254_

Round 1
Reviewer 1 Report
Comments and Suggestions for Authors
The manuscript demonstrates an exemplary level of scientific analysis based on the combination of two physical analytical techniques: Raman spectroscopy and GC–MS. The former provides direct characterization of structural and functional groups without the need for sample preparation, while the latter enables accurate and reliable identification of volatile constituents that form the basis of essential oils. This combined approach allowed the authors to achieve a deep interspecific differentiation and to identify unique compounds, including the first reported occurrence of cis,cis-nepetalactone in the essential oil of Minthostachys mollis. The study holds high scientific and practical significance. Nevertheless, in order to enhance methodological transparency and ensure reproducibility of the results, I would like to suggest several clarifications that, in my opinion, will be beneficial to both the authors and the broader scientific community.
- Figure 1 is comprehensively described in section 2.1. Analysis of Raman Spectral Data. However, to improve visual clarity and facilitate interpretation for readers, it would be beneficial to directly indicate the main bands on the spectra (e.g., using arrows or other markers) and include brief labels specifying the corresponding vibrational modes (such as C–H, C=O, C=C, etc.). This would allow readers to more easily correlate the graphical representation with the textual description and would enhance the overall interpretative value of the figure.
- A. citriodora is used in the caption of Figure S2, while the figure itself indicates A. citrodora. Both spellings may be acceptable, but it would be preferable to maintain a consistent nomenclature throughout the manuscript.
- The label “Data” on the y-axis in Figure 1 and in Figure S2 (Supplementary) is not informative, and a more descriptive label would be preferable. Additionally, the spectra are overlaid, but it is not clear whether the intensities were normalized or the raw spectra are shown — please clarify this in the figure caption or the text.
- In subsection 2.1, which is dedicated to the analysis of Raman spectra, it would be more appropriate to refer only to the averaged spectra (Figure S2.b) as an extension of Figure 1, since both figures present the original spectral data. In contrast, the PC1 loadings (Figure S2.a) represent the results of chemometric analysis and would more logically be introduced in the subsequent section where PCA is discussed.
- As far as I understood, the GC–MS data presented in Table 1 and subsequently used in the PCA and HCA analyses are based on normalized peak areas, which represent semi-quantitative information rather than absolute concentrations. However, this is not explicitly stated in the manuscript, which may lead to misinterpretation of the reported values as true mass or percentage composition. To enhance methodological transparency and avoid ambiguity, I recommend the following clarifications:
-
- Table 1. Please specify in the table caption that the values correspond to relative peak areas (%) derived from GC–MS signal normalization. Since this approach does not account for individual response factors, the values should not be interpreted as absolute or gravimetric percentages.
- PCA and HCA analysis. It would be important to indicate that the multivariate analysis was performed using semi-quantitative GC–MS data (normalized peak areas), rather than absolute concentration values. This clarification should be added either in the Materials and Methods section or in the caption of Figure 4.
- Terminology. To prevent misunderstanding, it is recommended to replace terms such as “composition (%)” or “chemical contribution” with more precise expressions, e.g. “relative peak area (%)” or “semi-quantitative contribution based on GC–MS signal intensity”.
-
- As far as I understood, in section 4.4 (lines 549–550) the authors state that “compound identification was further confirmed by injecting a homologous series of n-alkane standards (C8–C20).” However, the manuscript does not provide details about this procedure. It would be important to clarify what type of standard mixture was used (e.g., commercial solution or laboratory-prepared mixture), in what solvent and proportion it was prepared, the volume that was injected, and whether the injection and chromatographic conditions were identical to those applied for the analysis of the essential oil samples.
Author Response
The manuscript demonstrates an exemplary level of scientific analysis based on the combination of two physical analytical techniques: Raman spectroscopy and GC–MS. The former provides direct characterization of structural and functional groups without the need for sample preparation, while the latter enables accurate and reliable identification of volatile constituents that form the basis of essential oils. This combined approach allowed the authors to achieve a deep interspecific differentiation and to identify unique compounds, including the first reported occurrence of cis,cis-nepetalactone in the essential oil of Minthostachys mollis. The study holds high scientific and practical significance. Nevertheless, in order to enhance methodological transparency and ensure reproducibility of the results, I would like to suggest several clarifications that, in my opinion, will be beneficial to both the authors and the broader scientific community.
Response: Thank you for the comments. We added the changes and improved the manuscript.
1. Figure 1 is comprehensively described in section 2.1. Analysis of Raman Spectral Data. However, to improve visual clarity and facilitate interpretation for readers, it would be beneficial to directly indicate the main bands on the spectra (e.g., using arrows or other markers) and include brief labels specifying the corresponding vibrational modes (such as C–H, C=O, C=C, etc.). This would allow readers to more easily correlate the graphical representation with the textual description and would enhance the overall interpretative value of the figure.
Response: Thank you for the suggestion. We added the corresponding vibrational modes [Figure 1, 95-98]
2. A. citriodora is used in the caption of Figure S2, while the figure itself indicates A. citrodora. Both spellings may be acceptable, but it would be preferable to maintain a consistent nomenclature throughout the manuscript.
Response: Sorry for the mistake. We reviewed the name [Figure S2]
3. The label “Data” on the y-axis in Figure 1 and in Figure S2 (Supplementary) is not informative, and a more descriptive label would be preferable. Additionally, the spectra are overlaid, but it is not clear whether the intensities were normalized or the raw spectra are shown — please clarify this in the figure caption or the text.
Response: Thank for the suggestion. We added the label on the y-axis [Figure 1 and Figure S2]
4. In subsection 2.1, which is dedicated to the analysis of Raman spectra, it would be more appropriate to refer only to the averaged spectra (Figure S2.b) as an extension of Figure 1, since both figures present the original spectral data. In contrast, the PC1 loadings (Figure S2.a) represent the results of chemometric analysis and would more logically be introduced in the subsequent section where PCA is discussed.
Response: Figure S2a has been removed from the supplementary material. Supplementary Figure S2a now includes only the average Raman spectra of all samples, which are cited in subsection 2.1 as an extension of Figure 1.
5. As far as I understood, the GC–MS data presented in Table 1 and subsequently used in the PCA and HCA analyses are based on normalized peak areas, which represent semi-quantitative information rather than absolute concentrations. However, this is not explicitly stated in the manuscript, which may lead to misinterpretation of the reported values as true mass or percentage composition. To enhance methodological transparency and avoid ambiguity, I recommend the following clarifications:
- Table 1. Please specify in the table caption that the values correspond to relative peak areas (%) derived from GC–MS signal normalization. Since this approach does not account for individual response factors, the values should not be interpreted as absolute or gravimetric percentages.
Response: Thank for the comment. We reviewed the information in Table 1 [L. 247-250]
- PCA and HCA analysis. It would be important to indicate that the multivariate analysis was performed using semi-quantitative GC–MS data (normalized peak areas), rather than absolute concentration values. This clarification should be added either in the Materials and Methods section or in the caption of Figure 4.
Response: Thank for the suggestion. We incorporated the recommendations [L. 217-222, 228, 231-234, 243-244, 247, 249-250, 456-461, 574-582]
- To prevent misunderstanding, it is recommended to replace terms such as “composition (%)” or “chemical contribution” with more precise expressions, e.g. “relative peak area (%)” or “semi-quantitative contribution based on GC–MS signal intensity”.
Response: Thank. We replaced terms [L. 231-234, 237, 240, 241, 243, 246, 247, 249-250, 251, 257, 261, 272, 274, 295, 298, 302, 312, 321, 326, 329, 334, 340, 358, 366, 380, 383, 396, 399, 435, 471, 489-490]
6. As far as I understood, in section 4.4 (lines 549–550) the authors state that “compound identification was further confirmed by injecting a homologous series of n-alkane standards (C8–C20).” However, the manuscript does not provide details about this procedure. It would be important to clarify what type of standard mixture was used (e.g., commercial solution or laboratory-prepared mixture), in what solvent and proportion it was prepared, the volume that was injected, and whether the injection and chromatographic conditions were identical to those applied for the analysis of the essential oil samples.
Response: Thank you for the comment. We clarified the method [L. 574-582]
Reviewer 2 Report
Comments and Suggestions for Authors
In current study, eight different essential oils were characterized by the combination of Raman spectroscopy and GC-MS. A total of 224 components were identified. The essential oil of Chamaemelum nobile could not be identified duet to its low extraction yield and the high viscosity of the obtained concentrate. In my opinion, it is an interesting topic. However, I do not recommend to publish the at current stage. The following comments are to be carefully addressed:
- In Figure 2 and Figure 4, Hierarchical Classification of Plant Samples for the two methods were quite different. Based on the current result, Raman spectroscopy could not have the similar classification as GC-MS. Thus, the two results could not be combined. Meanwhile, the plant name in the two figures is different.
- In line 49, “a.s.l”should have the full word in the first time and use abbreviated form afterward.
- In line 65, what is “y MNR”?
- In line 141, what is the meaning of “δCHâ‚‚/CH₃”?
- The format of the references is different. For example, some use full journal name, some use abbreviated journal name.
Author Response
In current study, eight different essential oils were characterized by the combination of Raman spectroscopy and GC-MS. A total of 224 components were identified. The essential oil of Chamaemelum nobile could not be identified duet to its low extraction yield and the high viscosity of the obtained concentrate. In my opinion, it is an interesting topic. However, I do not recommend to publish the at current stage.
Response: Thank you for the comment. We reviewed the manuscript according to suggestions.
The following comments are to be carefully addressed:
1. In Figure 2 and Figure 4, Hierarchical Classification of Plant Samples for the two methods were quite different. Based on the current result, Raman spectroscopy could not have the similar classification as GC-MS. Thus, the two results could not be combined. Meanwhile, the plant name in the two figures is different.
Response: The results of hierarchical analysis using Raman spectroscopy (Figure 2) and GC–MS (Figure 4) differs because both methods capture complementary information: Raman reflects general molecular and vibrational profiles, while GC–MS provides relative abundances of individual volatile compounds. Therefore, clustering patterns are not expected to be identical [L. 456-461]. Also, we standardized the names of plants.
2. In line 49, “a.s.l”should have the full word in the first time and use abbreviated form afterward.
Response: We added the information [L. 49-50]
3. In line 65, what is “y MNR”?
Response: Sorry for the mistake. We changed the term [L. 66-67]
4. In line 141, what is the meaning of “δCHâ‚‚/CH₃”?
Response: Thank. We added the information [L. 140-141]
5. The format of the references is different. For example, some use full journal name, some use abbreviated journal name.
Response: We reviewed and corrected the information on the references section.
In addition, we noticed that Figure 5 and Figure S3 are missing from your article. Please check if they are missing, and if so, please add them.
Response: We reviewed the information [L. 516, 628-631]